# Semi-supervised Convolutional Neural Networks for Text Categorization via Region Embedding

**Rie Johnson**
RJ Research Consulting
Tarrytown, NY, USA
riejohnson@gmail.com

**Tong Zhang**[*]
Baidu Inc., Beijing, China
Rutgers University, Piscataway, NJ, USA
tzhang@stat.rutgers.edu

## Abstract

This paper presents a new semi-supervised framework with convolutional neural networks (CNNs) for text categorization. Unlike the previous approaches that rely on word embeddings, our method learns embeddings of small text regions from unlabeled data for integration into a supervised CNN. The proposed scheme for embedding learning is based on the idea of two-view semi-supervised learning, which is intended to be useful for the task of interest even though the training is done on unlabeled data. Our models achieve better results than previous approaches on sentiment classification and topic classification tasks.

## 1 Introduction

Convolutional neural networks (CNNs) [15] are neural networks that can make use of the internal structure of data such as the *2D structure* of image data through convolution layers, where each computation unit responds to a small region of input data (e.g., a small square of a large image). On text, CNN has been gaining attention, used in systems for tagging, entity search, sentence modeling, and so on [4, 5, 26, 7, 21, 12, 25, 22, 24, 13], to make use of the *1D structure* (word order) of text data. Since CNN was originally developed for image data, which is fixed-sized, low-dimensional and dense, without modification it cannot be applied to text documents, which are variable-sized, high-dimensional and sparse if represented by sequences of one-hot vectors. In many of the CNN studies on text, therefore, words in sentences are first converted to low-dimensional *word vectors*. The word vectors are often obtained by some other method from an additional large corpus, which is typically done in a fashion similar to language modeling though there are many variations [3, 4, 20, 23, 6, 19].

Use of word vectors obtained this way is a form of semi-supervised learning and leaves us with the following questions. Q1. How effective is CNN on text in a purely supervised setting without the aid of unlabeled data? Q2. Can we use unlabeled data with CNN more effectively than using general word vector learning methods? Our recent study [11] addressed Q1 on text categorization and showed that CNN without a word vector layer is not only feasible but also beneficial when not aided by unlabeled data. Here we address Q2 also on text categorization: building on [11], we propose a new semi-supervised framework that learns embeddings of small text *regions* (instead of *words*) from unlabeled data, for use in a supervised CNN.

The essence of CNN, as described later, is to convert small regions of data (e.g., "love it" in a document) to feature vectors for use in the upper layers; in other words, through training, a convolution layer learns an *embedding* of small regions of data. Here we use the term 'embedding' loosely to mean a structure-preserving function, in particular, a function that generates low-dimensional features that preserve the predictive structure. [11] applies CNN *directly to high-dimensional one-hot vectors*, which leads to *directly* learning an *embedding* of *small text regions* (e.g., regions of size 3

---
[*]Tong Zhang would like to acknowledge NSF IIS-1250985, NSF IIS-1407939, and NIH R01AI116744 for supporting his research.

like phrases, or regions of size 20 like sentences), eliminating the extra layer for word vector conversion. This direct learning of region embedding was noted to have the merit of higher accuracy with a simpler system (no need to tune hyper-parameters for word vectors) than supervised word vector-based CNN in which word vectors are randomly initialized and trained as part of CNN training. Moreover, the performance of [11]'s best CNN rivaled or exceeded the previous best results on the benchmark datasets.

Motivated by this finding, we seek effective use of unlabeled data for text categorization through *direct learning of embeddings of text regions*. Our new semi-supervised framework learns a *region embedding* from *unlabeled data* and uses it to produce additional input (additional to one-hot vectors) to supervised CNN, where a *region embedding* is trained with *labeled data*. Specifically, from unlabeled data, we learn *tv-embeddings* ('tv' stands for 'two-view'; defined later) of a text region through the task of predicting its surrounding context. According to our theoretical finding, a *tv-embedding* has desirable properties under ideal conditions on the relations between two views and the labels. While in reality the ideal conditions may not be perfectly met, we consider them as guidance in designing the tasks for tv-embedding learning.

We consider several types of tv-embedding learning task trained on unlabeled data; e.g., one task is to predict the presence of the concepts relevant to the intended task (e.g., 'desire to recommend the product') in the context, and we indirectly use labeled data to set up this task. Thus, we seek to learn tv-embeddings *useful specifically for the task of interest*. This is in contrast to the previous word vector/embedding learning methods, which typically produce a word embedding for general purposes so that all aspects (e.g., either syntactic or semantic) of words are captured. In a sense, the goal of our region embedding learning is to map text regions to high-level concepts relevant to the task. This cannot be done by word embedding learning since individual words in isolation are too primitive to correspond to high-level concepts. For example, "easy to use" conveys positive sentiment, but "use" in isolation does not. We show that our models with tv-embeddings outperform the previous best results on sentiment classification and topic classification. Moreover, a more direct comparison confirms that our region tv-embeddings provide more *compact and effective* representations of regions for the task of interest than what can be obtained by manipulation of a word embedding.

## 1.1   Preliminary: one-hot CNN for text categorization [11]

A CNN is a feed-forward network equipped with convolution layers interleaved with pooling layers. A convolution layer consists of computation units, each of which responds to a small region of input (e.g., a small square of an image), and the small regions collectively cover the entire data. A computation unit associated with the $\ell$-th region of input $\mathbf{x}$ computes:

$$\boldsymbol{\sigma}(\mathbf{W} \cdot \mathbf{r}_\ell(\mathbf{x}) + \mathbf{b}) , \tag{1}$$

where $\mathbf{r}_\ell(\mathbf{x}) \in \mathbb{R}^q$ is the input *region vector* that represents the $\ell$-th region. Weight matrix $\mathbf{W} \in \mathbb{R}^{m \times q}$ and bias vector $\mathbf{b} \in \mathbb{R}^m$ are *shared* by all the units in the same layer, and they are learned through training. In [11], input $\mathbf{x}$ is a document represented by one-hot vectors (Figure 1); therefore, we call [11]'s CNN *one-hot CNN*; $\mathbf{r}_\ell(\mathbf{x})$ can be either a concatenation of one-hot vectors, a bag-of-word vector (bow), or a bag-of-$n$-gram vector: e.g., for a region "love it"

$$\mathbf{r}_\ell(\mathbf{x}) = [\ \begin{matrix} \text{I} & \text{it} & \textbf{love} \\ 0 & 0 & 1 \end{matrix} \ | \ \begin{matrix} \text{I} & \textbf{it} & \text{love} \\ 0 & 1 & 0 \end{matrix} \ ]^\top \quad \text{(concatenation)} \tag{2}$$

$$\mathbf{r}_\ell(\mathbf{x}) = [\ \begin{matrix} \text{I} & \textbf{it} & \textbf{love} \\ 0 & 1 & 1 \end{matrix} \ ]^\top \quad \text{(bow)} \tag{3}$$

The bow representation (3) loses word order within the region but is more robust to data sparsity, enables a large region size such as 20, and speeds up training by having fewer parameters. This is what we mainly use for embedding learning from unlabeled data. CNN with (2) is called *seq-CNN* and CNN with (3) *bow-CNN*. The region size and stride (distance between the region centers) are meta-parameters. Note that we used a tiny three-word vocabulary for the vector examples above to save space, but a vocabulary of typical applications could be much larger. $\boldsymbol{\sigma}$ in (1) is a component-wise non-linear function (e.g., applying $\sigma(x) = \max(x, 0)$ to each vector component). Thus, each computation unit generates an $m$-dimensional vector where $m$ is the number of weight vectors ($\mathbf{W}$'s rows) or *neurons*. In other words, *a convolution layer embodies an embedding of text regions*, which produces an $m$-dim vector for each text region. In essence, a region embedding uses co-presence and absence of words in a region as input to produce predictive features, e.g., if presence of "easy

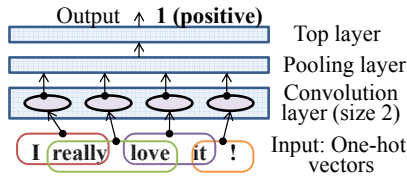

Figure 1: One-hot CNN example. Region size 2, stride 1.

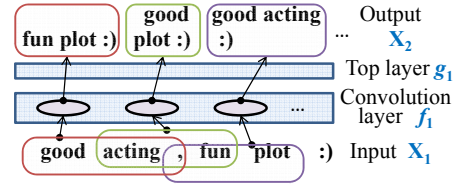

Figure 2: Tv-embedding learning by training to predict adjacent regions.

to use" with absence of "not" is a predictive indicator, it can be turned into a large feature value by having a negative weight on "not" (to penalize its presence) and positive weights on the other three words in one row of $\mathbf{W}$. A more formal argument can be found in the supplementary material. The $m$-dim vectors from all the text regions of each document are aggregated by the pooling layer, by either component-wise maximum (*max-pooling*) or average (*average-pooling*), and used by the top layer (a linear classifier) as features for classification. Here we focused on the convolution layer; for other details, [11] should be consulted.

## 2 Semi-supervised CNN with tv-embeddings for text categorization

It was shown in [11] that one-hot CNN is effective on text categorization, where the essence is direct learning of an embedding of text regions aided by new options of input region vector representation. We go further along this line and propose a semi-supervised learning framework that learns an embedding of text regions from unlabeled data and then integrates the learned embedding in supervised training. The first step is to learn an embedding with the following property.

**Definition 1** (tv-embedding). *A function $f_1$ is a tv-embedding of $\mathcal{X}_1$ w.r.t. $\mathcal{X}_2$ if there exists a function $g_1$ such that $P(X_2|X_1) = g_1(f_1(X_1), X_2)$ for any $(X_1, X_2) \in \mathcal{X}_1 \times \mathcal{X}_2$.*

A *tv-embedding* ('tv' stands for two-view) of a *view* ($X_1$), by definition, preserves everything required to predict another view ($X_2$), and it can be trained on unlabeled data. The motivation of tv-embedding is our theoretical finding (formalized in the Appendix) that, essentially, a tv-embedded feature vector $f_1(X_1)$ is as useful as $X_1$ for the purpose of classification *under ideal conditions*. The conditions essentially state that there exists a set $H$ of hidden concepts such that two views and labels of the classification task are related to each other *only* through the concepts in $H$. The concepts in $H$ might be, for example, "pricey", "handy", "hard to use", and so on for sentiment classification of product reviews. While in reality the ideal conditions may not be completely met, we consider them as guidance and design tv-embedding learning accordingly.

Tv-embedding learning is related to two-view feature learning [2] and ASO [1], which learn a linear embedding from unlabeled data through tasks such as predicting a word (or predicted labels) from the features associated with its surrounding words. These studies were, however, limited to a linear embedding. A related method in [6] learns a word embedding so that left context and right context maximally correlate in terms of canonical correlation analysis. While we share with these studies the general idea of using the relations of two views, we focus on nonlinear learning of region embeddings useful for the task of interest, and the resulting methods are very different. An important difference of tv-embedding learning from co-training is that it does not involve label guessing, thus avoiding risk of label contamination. [8] used a Stacked Denoising Auto-encoder to extract features invariant across domains for sentiment classification from unlabeled data. It is for fully-connected neural networks, which underperformed CNNs in [11].

Now let $\mathcal{B}$ be the base CNN model for the task of interest, and assume that $\mathcal{B}$ has one convolution layer with region size $p$. Note, however, that the restriction of having only one convolution layer is merely for simplifying the description. We propose a semi-supervised framework with the following two steps.

1. *Tv-embedding learning*: Train a neural network $\mathcal{U}$ to predict the context from each region of size $p$ so that $\mathcal{U}$'s convolution layer generates feature vectors for each text region of size $p$ for use in the classifier in the top layer. It is this convolution layer, which embodies the tv-embedding, that we transfer to the supervised learning model in the next step. (Note that $\mathcal{U}$ differs from CNN in that each small region is associated with its own target/output.)

2. *Final supervised learning*: Integrate the learned tv-embedding (the convolution layer of $\mathcal{U}$) into $\mathcal{B}$, so that the tv-embedded regions (the output of $\mathcal{U}$'s convolution layer) are used as an additional input to $\mathcal{B}$'s convolution layer. Train this final model with labeled data.

These two steps are described in more detail in the next two sections.

## 2.1 Learning tv-embeddings from unlabeled data

We create a task on unlabeled data to predict the context (adjacent text regions) from each region of size $p$ defined in $\mathcal{B}$'s convolution layer. To see the correspondence to the definition of tv-embeddings, it helps to consider a sub-task that assigns a label (e.g., positive/negative) to each text region (e.g., ", fun plot") instead of the ultimate task of categorizing the entire document. This is sensible because CNN makes predictions by building up from these small regions. In a document "good acting, fun plot :)" as in Figure 2, the clues for predicting a label of ", fun plot" are ", fun plot" itself (view-1: $X_1$) and its *context* "good acting" and ":)" (view-2: $X_2$). $\mathcal{U}$ is trained to predict $X_2$ from $X_1$, i.e., to approximate $P(X_2|X_1)$ by $g_1(f_1(X_1), X_2)$ as in Definition 1, and functions $f_1$ and $g_1$ are embodied by the convolution layer and the top layer, respectively.

Given a document $\mathbf{x}$, for each text region indexed by $\ell$, $\mathcal{U}$'s convolution layer computes:

$$\mathbf{u}_\ell(\mathbf{x}) = \boldsymbol{\sigma}^{(\mathcal{U})} \left( \mathbf{W}^{(\mathcal{U})} \cdot \mathbf{r}_\ell^{(\mathcal{U})}(\mathbf{x}) + \mathbf{b}^{(\mathcal{U})} \right), \tag{4}$$

which is the same as (1) except for the superscript "$(\mathcal{U})$" to indicate that these entities belong to $\mathcal{U}$. The top layer (a linear model for classification) uses $\mathbf{u}_\ell(\mathbf{x})$ as features for prediction. $\mathbf{W}^{(\mathcal{U})}$ and $\mathbf{b}^{(\mathcal{U})}$ (and the top-layer parameters) are learned through training. The input region vector representation $\mathbf{r}_\ell^{(\mathcal{U})}(\mathbf{x})$ can be either sequential, bow, or bag-of-$n$-gram, independent of $\mathbf{r}_\ell(\mathbf{x})$ in $\mathcal{B}$.

The goal here is to learn an embedding of text regions ($X_1$), shared with all the text regions at every location. Context ($X_2$) is used only in tv-embedding learning as prediction target (i.e., not transferred to the final model); thus, the representation of context should be determined to optimize the final outcome without worrying about the cost at prediction time. Our guidance is the conditions on the relationships between the two views mentioned above; ideally, the two views should be related to each other only through the relevant concepts. We consider the following two types of target/context representation.

**Unsupervised target** A straightforward vector encoding of context/target $X_2$ is bow vectors of the text regions on the left and right to $X_1$. If we distinguish the left and right, the target vector is $2|V|$-dimensional with vocabulary $V$, and if not, $|V|$-dimensional. One potential problem of this encoding is that adjacent regions often have syntactic relations (e.g., "the" is often followed by an adjective or a noun), which are typically irrelevant to the task (e.g., to identify positive/negative sentiment) and therefore undesirable. A simple remedy we found effective is *vocabulary control* of context to remove function words (or stop-words if available) from (and only from) the target vocabulary.

**Partially-supervised target** Another context representation that we consider is partially supervised in the sense that it uses labeled data. First, we train a CNN with the labeled data for the intended task and apply it to the unlabeled data. Then we discard the predictions and only retain the internal output of the convolution layer, which is an $m$-dimensional vector for each text region where $m$ is the number of neurons. We use these $m$-dimensional vectors to represent the context. [11] has shown, by examples, that each dimension of these vectors roughly represents concepts relevant to the task, e.g., 'desire to recommend the product', 'report of a faulty product', and so on. Therefore, an advantage of this representation is that there is no obvious noise between $X_1$ and $X_2$ since context $X_2$ is represented only by the concepts relevant to the task. A disadvantage is that it is only as good as the supervised CNN that produced it, which is not perfect and in particular, some relevant concepts would be missed if they did not appear in the labeled data.

## 2.2 Final supervised learning: integration of tv-embeddings into supervised CNN

We use the tv-embedding obtained from unlabeled data to produce *additional input* to $\mathcal{B}$'s convolution layer, by replacing $\boldsymbol{\sigma} \left( \mathbf{W} \cdot \mathbf{r}_\ell(\mathbf{x}) + \mathbf{b} \right)$ (1) with:

$$\boldsymbol{\sigma} \left( \mathbf{W} \cdot \mathbf{r}_\ell(\mathbf{x}) + \mathbf{V} \cdot \mathbf{u}_\ell(\mathbf{x}) + \mathbf{b} \right), \tag{5}$$

where $\mathbf{u}_\ell(\mathbf{x})$ is defined by (4), i.e., $\mathbf{u}_\ell(\mathbf{x})$ is the output of the tv-embedding applied to the $\ell$-th region. We train this model with the labeled data of the task; that is, we update the weights $\mathbf{W}$, $\mathbf{V}$, bias $\mathbf{b}$, and the top-layer parameters so that the designated loss function is minimized on the labeled training data. $\mathbf{W}^{(\mathcal{U})}$ and $\mathbf{b}^{(\mathcal{U})}$ can be either fixed or updated for fine-tuning, and in this work we fix them for simplicity.

Note that while (5) takes a tv-embedded region as input, (5) itself is also an embedding of text regions; let us call it (and also (1)) a *supervised embedding*, as it is trained with labeled data, to distinguish it from tv-embeddings. That is, we use tv-embeddings to improve the supervised embedding. Note that (5) can be naturally extended to accommodate multiple tv-embeddings by

$$\boldsymbol{\sigma}\left(\mathbf{W}\cdot\mathbf{r}_\ell(\mathbf{x})+\sum_{i=1}^{k}\mathbf{V}^{(i)}\cdot\mathbf{u}_\ell^{(i)}(\mathbf{x})+\mathbf{b}\right),\tag{6}$$

so that, for example, two types of tv-embedding (i.e., $k=2$) obtained with the unsupervised target and the partially-supervised target can be used at once, which can lead to performance improvement as they complement each other, as shown later.

## 3 Experiments

Our code and the experimental settings are available at `riejohnson.com/cnn_download.html`.

**Data**   We used the three datasets used in [11]: IMDB, Elec, and RCV1, as summarized in Table 1. IMDB (movie reviews) [17] comes with an unlabeled set. To facilitate comparison with previous studies, we used a union of this set and the training set as unlabeled data. Elec consists of Amazon reviews of electronics products. To use as unlabeled data, we chose 200K reviews from the same data source so that they are disjoint from the training and test sets, and that the reviewed products are disjoint from the test set. On the 55-way classification of the second-level topics on RCV1 (news), unlabeled data was chosen to be disjoint from the training and test sets. On the multi-label categorization of 103 topics on RCV1, since the official LYRL04 split for this task divides the entire corpus into a training set and a test set, we used the entire test set as unlabeled data (the transductive learning setting).

|       | #train | #test  | #unlabeled      | #class      | output            |
|-------|--------|--------|-----------------|-------------|-------------------|
| IMDB  | 25,000 | 25,000 | 75K (20M words) | 2           | Positive/negative |
| Elec  | 25,000 | 25,000 | 200K (24M words)| 2           | sentiment         |
| RCV1  | 15,564 | 49,838 | 669K (183M words)| 55 (single) | Topic(s)          |
|       | 23,149 | 781,265| 781K (214M words)| 103 (multi)† |                   |

Table 1: Datasets. †The multi-label RCV1 is used only in Table 6.

**Implementation**   We used the one-layer CNN models found to be effective in [11] as our base models $\mathcal{B}$, namely, seq-CNN on IMDB/Elec and bow-CNN on RCV1. Tv-embedding training minimized weighted square loss $\sum_{i,j}\alpha_{i,j}(\mathbf{z}_i[j]-\mathbf{p}_i[j])^2$ where $i$ goes through the regions, $\mathbf{z}$ represents the target regions, and $\mathbf{p}$ is the model output. The weights $\alpha_{i,j}$ were set to balance the loss originating from the presence and absence of words (or concepts in case of the partially-supervised target) and to speed up training by eliminating some negative examples, similar to negative sampling of [19]. To experiment with the unsupervised target, we set $\mathbf{z}$ to be bow vectors of adjacent regions on the left and right, while only retaining the 30K most frequent words with *vocabulary control*; on sentiment classification, function words were removed, and on topic classification, numbers and stop-words provided by [16] were removed. Note that these words were removed from (and only from) the target vocabulary. To produce the partially-supervised target, we first trained the supervised CNN models with 1000 neurons and applied the trained convolution layer to unlabeled data to generate 1000-dimensional vectors for each region. The rest of implementation follows [11]; i.e., supervised models minimized square loss with $L_2$ regularization and optional dropout [9]; $\boldsymbol{\sigma}$ and $\boldsymbol{\sigma}^{(\mathcal{U})}$ were the rectifier; response normalization was performed; optimization was done by SGD.

**Model selection**   On all the tested methods, tuning of meta-parameters was done by testing the models on the held-out portion of the training data, and then the models were re-trained with the chosen meta-parameters using the entire training data.

## 3.1 Performance results

**Overview** After confirming the effectiveness of our new models in comparison with the supervised CNN, we report the performances of [13]'s CNN, which relies on word vectors pre-trained with a very large corpus (Table 3). Besides comparing the performance of approaches as a whole, it is also of interest to compare the usefulness of what was learned from unlabeled data; therefore, we show how it performs if we integrate the word vectors into our base model one-hot CNNs (Figure 3). In these experiments we also test word vectors trained by word2vec [19] on our unlabeled data (Figure 4). We then compare our models with two standard semi-supervised methods, transductive SVM (TSVM) [10] and co-training (Table 3), and with the previous best results in the literature (Tables 4–6). In all comparisons, our models outperform the others. In particular, our region tv-embeddings are shown to be more compact and effective than region embeddings obtained by simple manipulation of word embeddings, which supports our approach of using region embedding instead of word embedding.

| names in Table 3 | $X_1$: $\mathbf{r}_\ell^{(\mathcal{U})}(\mathbf{x})$ | $X_2$: target of $\mathcal{U}$ training |
|---|---|---|
| unsup-tv. | bow vector | bow vector |
| parsup-tv. | bow vector | output of supervised embedding |
| unsup3-tv. | bag-of-{1,2,3}-gram vector | bow vector |

Table 2: Tested tv-embeddings.

| | | | IMDB | Elec | RCV1 |
|---|---|---|---|---|---|
| 1 | linear SVM with 1-3grams [11] | | 10.14 | 9.16 | 10.68 |
| 2 | linear TSVM with 1-3grams | | 9.99 | 16.41 | 10.77 |
| 3 | [13]'s CNN | | 9.17 | 8.03 | 10.44 |
| 4 | One-hot CNN (simple) [11] | | 8.39 | 7.64 | 9.17 |
| 5 | One-hot CNN (simple) co-training best | | (8.06) | (7.63) | (8.73) |
| 6 | | 100-dim | 7.12 | 6.96 | 8.10 |
| 7 | unsup-tv. | 200-dim | 6.81 | 6.69 | 7.97 |
| 8 | | 100-dim | 7.12 | 6.58 | 8.19 |
| 9 | Our CNN parsup-tv. | 200-dim | 7.13 | 6.57 | 7.99 |
| 10 | | 100-dim | 7.05 | 6.66 | 8.13 |
| 11 | unsup3-tv. | 200-dim | 6.96 | 6.84 | 8.02 |
| 12 | all three | 100×3 | **6.51** | **6.27** | **7.71** |

Table 3: Error rates (%). For comparison, all the CNN models were constrained to have 1000 neurons. The parentheses around the error rates indicate that co-training meta-parameters were tuned on test data.

**Our CNN with tv-embeddings** We tested three types of tv-embedding as summarized in Table 2. The first thing to note is that all of our CNNs (Table 3, row 6–12) outperform their supervised counterpart in row 4. This confirms the effectiveness of the framework we propose. In Table 3, for meaningful comparison, all the CNNs are constrained to have exactly one convolution layer (except for [13]'s CNN) with 1000 neurons. The best-performing supervised CNNs within these constraints (row 4) are: seq-CNN (region size 3) on IMDB and Elec and bow-CNN (region size 20) on RCV1[1]. They also served as our base models $\mathcal{B}$ (with region size parameterized on IMDB/Elec). More complex supervised CNNs from [11] will be reviewed later. On sentiment classification (IMDB and Elec), the region size chosen by model selection for our models was 5, larger than 3 for the supervised CNN. This indicates that unlabeled data enabled effective use of larger regions which are more predictive but might suffer from data sparsity in supervised settings.

'unsup3-tv.' (rows 10–11) uses a bag-of-$n$-gram vector to initially represent each region, thus, retains word order partially within the region. When used individually, unsup3-tv. did not outperform the other tv-embeddings, which use bow instead (rows 6–9). But we found that it contributed to error reduction when combined with the others (not shown in the table). This implies that it learned from unlabeled data predictive information that the other two embeddings missed. The best performances (row 12) were obtained by using all the three types of tv-embeddings at once according to (6). By doing so, the error rates were improved by nearly 1.9% (IMDB) and 1.4% (Elec and RCV1) compared with the supervised CNN (row 4), as a result of the three tv-embeddings with different strengths complementing each other.

|      | concat | avg  |
|------|--------|------|
| IMDB | 8.31   | 7.83 |
| Elec | 7.37   | 7.24 |
| RCV1 | 8.70   | 8.62 |

Figure 3: GN word vectors integrated into our base models. Better than [13]'s CNN (Table 3, row 3).

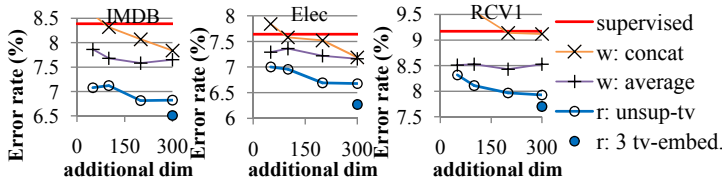

Figure 4: Region tv-embeddings vs. word2vec word embeddings. Trained on our unlabeled data. $x$-axis: dimensionality of the additional input to supervised region embedding. 'r:': region, 'w:': word.

**[13]'s CNN** It was shown in [13] that CNN that uses the Google News word vectors as input is competitive on a number of sentence classification tasks. These vectors (300-dimensional) were trained by the authors of word2vec [19] on a very large Google News (*GN*) corpus (100 billion words; 500–5K times larger than our unlabeled data). [13] argued that these vectors can be useful for various tasks, serving as 'universal feature extractors'. We tested [13]'s CNN, which is equipped with three convolution layers with different region sizes (3, 4, and 5) and max-pooling, using the GN vectors as input. Although [13] used only 100 neurons for each layer, we changed it to 400, 300, and 300 to match the other models, which use 1000 neurons. Our models clearly outperform these models (Table 3, row 3) with relatively large differences.

**Comparison of embeddings** Besides comparing the performance of the approaches as a whole, it is also of interest to compare the usefulness of what was learned from unlabeled data. For this purpose, we experimented with integration of a word embedding into our base models using two methods; one takes the concatenation, and the other takes the average, of word vectors for the words in the region. These provide additional input to the supervised embedding of regions in place of $\mathbf{u}_\ell(\mathbf{x})$ in (5). That is, for comparison, we produce a region embedding from a word embedding to replace a region tv-embedding. We show the results with two types of word embeddings: the GN word embedding above (Figure 3), and word embeddings that we trained with the word2vec software on our unlabeled data, i.e., the same data as used for tv-embedding learning and all others (Figure 4). Note that Figure 4 plots error rates in relation to the dimensionality of the produced additional input; a smaller dimensionality has an advantage of faster training/prediction.

On the results, first, the region tv-embedding is more useful for these tasks than the tested word embeddings since the models with a tv-embedding clearly outperform all the models with a word embedding. Word vector concatenations of much higher dimensionality than those shown in the figure still underperformed 100-dim region tv-embedding. Second, since our region tv-embedding takes the form of $\boldsymbol{\sigma}(\mathbf{W} \cdot \mathbf{r}_\ell(\mathbf{x}) + \mathbf{b})$ with $\mathbf{r}_\ell(\mathbf{x})$ being a bow vector, the columns of $\mathbf{W}$ correspond to words, and therefore, $\mathbf{W} \cdot \mathbf{r}_\ell(\mathbf{x})$ is the sum of $\mathbf{W}$'s columns whose corresponding words are in the $\ell$-th region. Based on that, one might wonder why we should not simply use the sum or average of word vectors obtained by an existing tool such as word2vec instead. The suboptimal performances of 'w: average' (Figure 4) tells us that this is a bad idea. We attribute it to the fact that region embeddings learn predictiveness of co-presence and absence of words in a region; a region embedding can be more expressive than averaging of word vectors. Thus, an *effective* and *compact* region embedding cannot be trivially obtained from a word embedding. In particular, effectiveness of the combination of three tv-embeddings ('r: 3 tv-embed.' in Figure 4) stands out.

Additionally, our mechanism of using information from unlabeled data is more effective than [13]'s CNN since our CNNs with GN (Figure 3) outperform [13]'s CNNs with GN (Table 3, row 3). This is because in our model, one-hot vectors (the original features) compensate for potential information loss in the embedding learned from unlabeled data. This, as well as region-vs-word embedding, is a major difference between our model and [13]'s model.

**Standard semi-supervised methods** Many of the standard semi-supervised methods are not applicable to CNN as they require bow vectors as input. We tested TSVM with bag-of-{1,2,3}-gram vectors using SVMlight. TSVM underperformed the supervised SVM[2] on two of the three datasets

| | | | | | |
|---|---|---|---|---|---|
| NB-LM 1-3grams [18] | 8.13 | – | SVM 1-3grams [11] | 8.71 | – |
| [11]'s best CNN | 7.67 | – | dense NN 1-3grams [11] | 8.48 | – |
| Paragraph vectors [14] | 7.46 | Unlab.data | NB-LM 1-3grams [11] | 8.11 | – |
| Ensemble of 3 models [18] | 7.43 | Ens.+unlab. | [11]'s best CNN | 7.14 | – |
| Our best | **6.51** | Unlab.data | Our best | **6.27** | Unlab.data |

Table 4:  IMDB: previous error rates (%).    Table 5:  Elec: previous error rates (%).

| models | micro-F | macro-F | extra resource |
|---|---|---|---|
| SVM [16] | 81.6 | 60.7 | – |
| bow-CNN [11] | 84.0 | 64.8 | – |
| bow-CNN w/ three tv-embed. | **85.7** | **67.1** | Unlabeled data |

Table 6:  RCV1 micro- and macro-averaged F on the multi-label task (103 topics) with the LYRL04 split.

(Table 3, rows 1–2). Since co-training is a meta-learner, it can be used with CNN. Random split of vocabulary and split into the first and last half of each document were tested. To reduce the computational burden, we report the best (and unrealistic) co-training performances obtained by optimizing the meta-parameters including when to stop *on the test data*. Even with this unfair advantage to co-training, co-training (Table 3, row 5) clearly underperformed our models. The results demonstrate the difficulty of effectively using unlabeled data on these tasks, given that the size of the labeled data is relatively large.

**Comparison with the previous best results**   We compare our models with the previous best results on IMDB (Table 4). Our best model with three tv-embeddings outperforms the previous best results by nearly 0.9%. All of our models with a single tv-embed. (Table 3, row 6–11) also perform better than the previous results. Since Elec is a relatively new dataset, we are not aware of any previous semi-supervised results. Our performance is better than [11]'s best supervised CNN, which has a complex network architecture of three convolution-pooling pairs in parallel (Table 5). To compare with the benchmark results in [16], we tested our model on the multi-label task with the LYRL04 split [16] on RCV1, in which more than one out of 103 categories can be assigned to each document. Our model outperforms the best SVM of [16] and the best supervised CNN of [11] (Table 6).

## 4   Conclusion

This paper proposed a new semi-supervised CNN framework for text categorization that learns embeddings of text regions with unlabeled data and then labeled data. As discussed in Section 1.1, a region embedding is trained to learn the predictiveness of co-presence and absence of words in a region. In contrast, a word embedding is trained to only represent individual words in isolation. Thus, a region embedding can be more expressive than simple averaging of word vectors in spite of their seeming similarity. Our comparison of embeddings confirmed its advantage; our region tv-embeddings, which are trained specifically for the task of interest, are more effective than the tested word embeddings. Using our new models, we were able to achieve higher performances than the previous studies on sentiment classification and topic classification.

## Appendix A   Theory of tv-embedding

Suppose that we observe two views $(X_1, X_2) \in \mathcal{X}_1 \times \mathcal{X}_2$ of the input, and a target label $Y \in \mathcal{Y}$ of interest, where $\mathcal{X}_1$ and $\mathcal{X}_2$ are finite discrete sets.

**Assumption 1.**   *Assume that there exists a set of hidden states $\mathcal{H}$ such that $X_1$, $X_2$, and $Y$ are conditionally independent given $h$ in $\mathcal{H}$, and that the rank of matrix $[P(X_1, X_2)]$ is $|\mathcal{H}|$.*

**Theorem 1.**   *Consider a tv-embedding $f_1$ of $\mathcal{X}_1$ w.r.t. $\mathcal{X}_2$. Under Assumption 1, there exists a function $q_1$ such that $P(Y|X_1) = q_1(f_1(X_1), Y)$. Further consider a tv-embedding $f_2$ of $\mathcal{X}_2$ w.r.t. $\mathcal{X}_1$. Then, under Assumption 1, there exists a function $q$ such that $P(Y|X_1, X_2) = q(f_1(X_1), f_2(X_2), Y)$.*

The proof can be found in the supplementary material.

## Footnotes

[1] The error rate on RCV1 in row 4 slightly differs from [11] because here we did not use the stopword list.

[2] Note that for feasibility, we only used the 30K most frequent $n$-grams in the TSVM experiments, thus, showing the SVM results also with 30K vocabulary for comparison, though on some datasets SVM performance can be improved by use of all the $n$-grams (e.g., 5 million $n$-grams on IMDB) [11]. This is because the computational cost of TSVM (single-core) turned out to be high, taking several days even with 30K vocabulary.

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
