[Supplementary Material]

# Supplementary material
# Semi-supervised Convolutional Neural Networks for Text Categorization via Region Embedding

**Rie Johnson**
RJ Research Consulting
Tarrytown, NY, USA
riejohnson@gmail.com

**Tong Zhang**
Baidu Inc., Beijing, China
Rutgers University, Piscataway, NJ, USA
tzhang@stat.rutgers.edu

This supplementary material contains the detailed proof of Theorem 1 in the main text, and some theoretical analysis concerning the representation power of region embedding to formally argue for its effectiveness.

## 1 Proof of Theorem 1

For completeness, the statement of the theorem as well as related assumptions and definitions are included.

Suppose that we observe two views $(X_1, X_2) \in \mathcal{X}_1 \times \mathcal{X}_2$ of the input, and a target label $Y \in \mathcal{Y}$ of interest, where $\mathcal{X}_1$ and $\mathcal{X}_2$ are finite discrete sets.

**Assumption 1.** *Assume that there exists a set of hidden states $\mathcal{H}$ such that $X_1$, $X_2$, and $Y$ are conditionally independent given $h$ in $\mathcal{H}$, and that the rank of matrix $[P(X_1, X_2)]$ is $|\mathcal{H}|$.*

**Definition 1** (multi-view embedding)**.** *A function $f_1$ is a multi-view embedding of $\mathcal{X}_1$ w.r.t. $\mathcal{X}_2$ if there exists a function $g_1$ such that $P(X_2|X_1) = g_1(f_1(X_1), X_2)$ for any $(X_1, X_2) \in \mathcal{X}_1 \times \mathcal{X}_2$.*

**Theorem 1.** *Consider a multi-view embedding $f_1$ of $\mathcal{X}_1$ w.r.t. $\mathcal{X}_2$. Under Assumption 1, there exists a function $q_1$ such that $P(Y|X_1) = q_1(f_1(X_1), Y)$.*
*Further consider a multi-view embedding $f_2$ of $\mathcal{X}_2$ w.r.t. $\mathcal{X}_1$. Then, under Assumption 1, there exists a function $q$ such that*

$$P(Y|X_1, X_2) = q(f_1(X_1), f_2(X_2), Y).$$

*Proof.* First, assume that $\mathcal{X}_1$ contains $d_1$ elements, and $\mathcal{X}_2$ contains $d_2$ elements, and $|\mathcal{H}| = k$. The independence and rank condition in Assumption 1 implies the decomposition

$$P(X_2|X_1) = \sum_{h \in \mathcal{H}} P(X_2|h)P(h|X_1)$$

is of rank $k$ if we consider $P(X_2|X_1)$ as a $d_2 \times d_1$ matrix (which we denote by $\mathbf{A}$). Now we may also regard $P(X_2|h)$ as a $d_2 \times k$ matrix (which we denote by $\mathbf{B}$), and $P(h|X_1)$ as a $k \times d_1$ matrix (which we denote by $\mathbf{C}$). From the matrix equation $\mathbf{A} = \mathbf{BC}$, we obtain $\mathbf{C} = (\mathbf{B}^\top \mathbf{B})^{-1}\mathbf{B}^\top \mathbf{A}$. Consider the $k \times d_2$ matrix $\mathbf{U} = (\mathbf{B}^\top \mathbf{B})^{-1}\mathbf{B}^\top$. Then we know that its elements correspond to a function of $(h, X_2) \in \mathcal{H} \times \mathcal{X}_2$. Therefore the relationship $\mathbf{C} = \mathbf{UA}$ implies that there exists a function $u(h, X_2)$ such that

$$\forall h \in \mathcal{H} : P(h|X_1) = \sum_{X_2 \in \mathcal{X}_2} P(X_2|X_1)u(h, X_2).$$

Using the definition of embedding in Definition 1, we obtain

$$P(h|X_1) = \sum_{X_2 \in \mathcal{X}_2} g_1(f_1(X_1), X_2)u(h, X_2).$$

Define $t_1(a_1, h) = \sum_{X_2} g_1(a_1, X_2)u(h, X_2)$, then for any $h \in \mathcal{H}$ we have

$$P(h|X_1) = t_1(f_1(X_1), h). \tag{1}$$

Similarly, there exists a function $t_2(a_2, h)$ such that for any $h \in \mathcal{H}$

$$P(h|X_2) = t_2(f_2(X_2), h). \tag{2}$$

Observe that

$$P(Y|X_1) = \sum_{h \in \mathcal{H}} P(Y, h|X_1) = \sum_{h \in \mathcal{H}} P(h|X_1)P(Y|h, X_1)$$

$$= \sum_{h \in \mathcal{H}} P(h|X_1)P(Y|h) = \sum_{h \in \mathcal{H}} t_1(f_1(X_1), h)P(Y|h)$$

where the third equation has used the assumption that $Y$ is independent of $X_1$ given $h$ and the last equation has used (1). By defining $q_1(a_1, Y) = \sum_{h \in \mathcal{H}} t_1(a_1, h)P(Y|h)$, we obtain $P(Y|X_1) = q_1(f_1(X_1), Y)$, as desired.

Further observe that

$$P(Y|X_1, X_2) = \sum_{h \in \mathcal{H}} P(Y, h|X_1, X_2)$$

$$= \sum_{h \in \mathcal{H}} P(h|X_1, X_2)P(Y|h, X_1, X_2)$$

$$= \sum_{h \in \mathcal{H}} P(h|X_1, X_2)P(Y|h), \tag{3}$$

where the last equation has used the assumption that $Y$ is independent of $X_1$ and $X_2$ given $h$.

Note that

$$P(h|X_1, X_2) = \frac{P(h, X_1, X_2)}{P(X_1, X_2)} = \frac{P(h, X_1, X_2)}{\sum_{h' \in \mathcal{H}} P(h', X_1, X_2)}$$

$$= \frac{P(h)P(X_1|h)P(X_2|h)}{\sum_{h' \in \mathcal{H}} P(h')P(X_1|h')P(X_2|h')}$$

$$= \frac{P(h, X_1)P(h, X_2)/P(h)}{\sum_{h' \in \mathcal{H}} P(h', X_1)P(h', X_2)/P(h')}$$

$$= \frac{P(h|X_1)P(h|X_2)/P(h)}{\sum_{h' \in \mathcal{H}} P(h'|X_1)P(h'|X_2)/P(h')}$$

$$= \frac{t_1(f_1(X_1), h)t_2(f_2(X_2), h)/P(h)}{\sum_{h' \in \mathcal{H}} t_1(f_1(X_1), h')t_2(f_2(X_2), h')/P(h')},$$

where the third equation has used the assumption that $X_1$ is independent of $X_2$ given $h$, and the last equation has used (1) and (2). The last equation means that $P(h|X_1, X_2)$ is a function of $(f_1(X_1), f_2(X_2), h)$. That is, there exists a function $\tilde{t}$ such that $P(h|X_1, X_2) = \tilde{t}(f_1(X_1), f_2(X_2), h)$. From (3), this implies that

$$P(Y|X_1, X_2) = \sum_{h \in \mathcal{H}} \tilde{t}(f_1(X_1), f_2(X_2), h)P(Y|h).$$

Now the theorem follows by defining $q(a_1, a_2, Y) = \sum_{h \in \mathcal{H}} \tilde{t}(a_1, a_2, h)P(Y|h)$. □

## 2 Representation Power of Region Embedding

In this section, we provide some formal definitions and theoretical arguments to support the effectiveness of the type of region embedding experimented with in the main text.

A text region embedding is a function that maps a region of text (a sequence of two or more words) into a numerical vector. The particular form of region embedding we consider takes either sequential

or bow representation of the text region as input. More precisely, consider a language with vocabulary $V$. Each word $w$ in the language is taken from $V$, and can be represented as a $|V|$ dimensional vector referred to as one-hot-vector representation. Each of the $|V|$ vector components represents a vocabulary entry. The vector representation of $w$ has value one for the component corresponding to the word, and value zeros elsewhere. A text region of size $m$ is a sequence of $m$ words $(w_1, w_2, \ldots, w_m)$, where each word $w_i \in V$. It can be represented as a $m|V|$ dimensional vector, which is a concatenation of vector representations of the words, as in (2) in Section 1.1 of the main text. Here we call this representation *seq-representation*. An alternative is the *bow-representation* as in (3) of the main text.

Let $\mathcal{R}_m$ be the set of all possible text regions of size $m$ in the seq-representation (or alternatively, bow-representation). We consider embeddings of a text region $\mathbf{x} \in \mathcal{R}_m$ in the form of

$$(\mathbf{Wx} + \mathbf{b})_+ = \max(0, \mathbf{Wx} + \mathbf{b}) .$$

The embedding matrix $\mathbf{W}$ and bias vector $\mathbf{b}$ are learned by training, and the training objective depends on the task. In the following, this particular form of region embedding is referred to as *RETEX* (Region Embedding of TEXt), and the vectors produced by RETEX or the results of RETEX are referred to as *RETEX vectors*.

The goal of region embedding learning is to map high-level concepts (relevant to the task of interest) to low-dimensional vectors. As said in the main text, this cannot be done by word embedding learning since a word embedding embeds individual words in isolation (i.e., word-$i$ is mapped to vector-$i$ irrespective of its context), which are too primitive to correspond to high-level concepts. For example, "easy to use" conveys positive sentiment, but "use" in isolation does not. Through the analysis of the representation power of RETEX, we show that unlike word embeddings, RETEX can model high-level concepts by using co-presence and absence of words in the region, which is similar to the traditional use of $m$-grams but more efficient/robust.

First we show that for any (possibly nonlinear) real-valued function $f(\cdot)$ defined on $\mathcal{R}_m$, there exists a RETEX so that this function can be expressed in terms of a linear function of RETEX vectors. This property is often referred to as *universal approximation* in the literature (e.g., see https://en.wikipedia.org/wiki/Universal_approximation_theorem).

**Proposition 1.** *Consider a real-valued function $f(\cdot)$ defined on $\mathcal{R}_m$. There exists an embedding matrix $\mathbf{W}$, bias vector $\mathbf{b}$, and vector $\mathbf{v}$ such that $f(\mathbf{x}) = \mathbf{v}^\top (\mathbf{Wx} + \mathbf{b})_+$ for all $\mathbf{x} \in \mathcal{R}_m$.*

*Proof.* Denote by $\mathbf{W}_{i,j}$ the entry of $\mathbf{W}$ corresponding to the $i$-th row and $j$-th column. Assume each element in $\mathcal{R}_m$ can be represented as a $d$ dimensional vector with no more than $m$ ones (and the remaining entries are zeros). Given a specific $\mathbf{x}^i \in \mathcal{R}_m$, let $S_i$ be a set of indexes $j \in \{1, \ldots, d\}$ such that the $j$-th component of $\mathbf{x}^i$ is one. We create a row $\mathbf{W}_{i,\cdot}$ in $\mathbf{W}$ such that $\mathbf{W}_{i,j} = 2I(j \in S_i) - 1$ for $1 \leq j \leq d$, where $I(\cdot)$ is the set indicator function. Let $\mathbf{b}_i = -|S_i| + 1$ where $\mathbf{b}_i$ denotes the $i$-th component of $\mathbf{b}$. It follows that $\mathbf{W}_{i,\cdot}\mathbf{x} + \mathbf{b}_i = 1$ if $\mathbf{x} = \mathbf{x}^i$, and $\mathbf{W}_{i,\cdot}\mathbf{x} + \mathbf{b}_i \leq 0$ otherwise. In this manner we create one row of $\mathbf{W}$ per every member of $\mathcal{R}_m$. Let $\mathbf{v}_i = f(\mathbf{x}^i)$. Then it follows that $f(\mathbf{x}) = \mathbf{v}^\top (\mathbf{Wx} + \mathbf{b})_+$. $\qquad\square$

The proof essentially constructs the indicator functions of all the $m$-grams (text regions of size $m$) in $\mathcal{R}_m$ and maps them to the corresponding function values. Thus, the representation power of RETEX is at least as good as $m$-grams, and more powerful than the sum of word embeddings in spite of the seeming similarity in form. However, it is well known that the traditional $m$-gram-based approaches, which assign one vector dimension per $m$-gram, can suffer from the data sparsity problem because an $m$-gram is useful only if it is seen in the training data.

This is where RETEX can have clear advantages. We show below that it can map similar $m$-grams (similar w.r.t. the training objective) to similar lower-dimensional vectors, which helps learning the task of interest. It is also more expressive than the traditional $m$-gram-based approaches because it can map not only co-presence but also absence of words (which $m$-gram cannot express concisely) into a single dimension. These properties lead to robustness to data sparsity.

We first introduce a definition of a *simple concept*.

**Definition 2.** *Consider $\mathcal{R}_m$ of the seq-representation. A high level semantic concept $C \subset \mathcal{R}_m$ is called simple if it can be defined as follows. Let $V_1, \ldots, V_m \subset V$ be $m$ word groups (each word*

*group may either represent a set of similar words or the absent of certain words), and $s_1, \ldots, s_m \subset \{\pm 1\}$ be signs. Define $C$ such that $\mathbf{x} \in C$ if and only if the $i$-th word in $\mathbf{x}$ either belongs to $V_i$ (if $s_i = 1$) or $\neg V_i$ (if $s_i = -1$).*

The next proposition illustrates the points above by stating that RETEX has the ability to represent a simple concept (defined above via the notion of similar words) by a single dimension. This is in contrast to the construction in the proof of Proposition 1, where one dimension could represent only one $m$-gram.

**Proposition 2.** *The indicator function of any simple concept $C$ can be embedded into one dimension using RETEX.*

*Proof.* Consider a text region vector $\mathbf{x} \in \mathcal{R}_m$ in seq-representation that contains $m$ of $|V|$-dimensional segments, where the $i$-th segment represents the $i$-th position in the text region. Let the $i$-th segment of $\mathbf{w}$ be a vector of zeros except for those components in $V_i$ being $s_i$. Let $b = 1 - \sum_{i=1}^{m} (s_i + 1)/2$. Then it is not difficult to check that $I(\mathbf{x} \in C) = (\mathbf{w}^\top \mathbf{x} + b)_+$. $\square$

The following proposition shows that RETEX can embed concepts that are unions of simple concepts into low-dimensional vectors.

**Proposition 3.** *If $C \subset \mathcal{R}_m$ is the union of $q$ simple concepts $C_1, \ldots, C_q$, then there exists a function $f(\mathbf{x})$ that is the linear function of $q$-dimensional RETEX vectors so that $\mathbf{x} \in C$ if and only if $f(\mathbf{x}) > 0$.*

*Proof.* Let $\mathbf{b} \in \mathbb{R}^q$, and let $\mathbf{W}$ have $q$ rows, so that $I(\mathbf{x} \in C_i) = (\mathbf{W}_{i,\cdot} \mathbf{x} + \mathbf{b}_i)_+$ for each row $i$, as constructed in the proof of Proposition 2. Let $\mathbf{v} = [1, \ldots, 1]^\top \in \mathbb{R}^q$. Then $f(\mathbf{x}) = \mathbf{v}^\top (\mathbf{W}\mathbf{x} + \mathbf{b})_+$ is a function of the desired property. $\square$

Note that $q$ can be much smaller than the number of $m$-grams in concept $C$. Proposition 3 shows that RETEX has the ability to simultaneously make use of word similarity (via word groups) and the fact that words occur in the context, to reduce the embedding dimension. A word embedding can model word similarity but does not model context. $m$-gram-based approaches can model context but cannot model word similarity — which means a concept/context has to be expressed with a large number of individual $m$-grams, leading to the data sparsity problem. Thus, the representation power of RETEX exceeds that of single-word embedding and traditional $m$-gram-based approaches.