[Reviews · NeurIPS 2015]

Submitted by Assigned_Reviewer_1

This paper presents an unsupervised embedding learning from two views, text region such as a three word phrase and its surrounding. A standard two step CNN is presented where first module applies an unsupervised embedding learning from unlabled data and the second one uses the previous embeddings as pre-trained embeddings on a supervised task. The two steps can use different datasets (hence the embedding learning step can totally be applied on unlabeled data)

The paper is very well written and motivation is clear. The benchmark analysis back up the presented approach.

The paper tries to show if semi-supervised learning,

where unsupervised embeddings are pre-trained for CNN is useful for CNN training as well as investigates a better way of learning these embeddings. The idea is to learn the context embeddings(output) from the region embeddings (input) simulating a multi-view learning approach. They prove that their embedding learning method outperforms supervised CNN, contrary to what have been reported in an earlier work (ref#10). What is presented as new approach in this paper is rather a change in the encoding of the data into the model, i.e., text regions (n-grams) rather than words, to train unsupervised embeddings which are then used as pre-trained embeddings to a supervised CNN.

Summary: An extension to an earlier published paper with a twist where embeddings are learnt on text regions (similar to images), the new embedding learning's performance is supported by experiments.

Submitted by Assigned_Reviewer_2

The paper presents an approach to learn features for region representation from unlabeled data, for use in text classification. The paper shows that using unlabeled data to estimate predictive features for text regions in a two-view learning framework brings substantial improvements over purely supervised learning of representations, and that learning region embeddings is substantially more effective than using word-embeddings (such as ones produced via word2vec).

The paper explores different learning targets from unlabeled data as well as different representations of regions (bag-of-words vs concatenation). It is shown that the resulting representations are complementary and the combination is very effective.

The main lesson from the paper: that learning region embeddings on unlabeled data is substantially more effective than the commonly employed method of using word-embeddings on unlabeled data and relying solely on them to represent regions, has substantial implications for NLP research.

The paper is written very well and is easy to follow. The approach is original. The experiments are very thorough and bring insight into the performance of the different options.

Improvements over the previous state-of-the-art obtained by use of deeper CNN models or

semi-supervised n-gram models are substantial and convincing.
Summary: This is a very well written paper which presents methods to learn text region representations from unlabeled data from unsupervised or partially-supervised two-view learning prediction tasks. The proposed method does substantially better than the commonly used approach of representing regions solely based on word embeddings trained from unlabeled data, and achieves convincing improvements in the state-of-the-art on benchmark datasets.

Submitted by Assigned_Reviewer_3

This is a nicely done paper concerning CNN's for text classification, and specifically a semi-supervised extension. I have only relatively minor comments.

In general, the motivation of considering the model from the "two-view" (tv) vantage point was unclear to me. This seems like a straight-forward instance of semi-supervised learning in the classical sense, and I did not really see the purpose of Definition 1. Indeed I think introducing the "two-view" concept was a bit distracting, especially because this seems rather different from standard "multi-view" learning in which ones learns jointly from distinct representations of instances. I would have appreciated a few words contrasting the proposed approach with co-training (I realize that the authors compare to this, of course, but a description of similarities and differences would be welcome).

A small comment on the model: the authors note that a constraint on the input regions to the unsupervised model (i.e., r^U_l(x)) is that it must have the same region size as B. But in Eq (5), an entirely separate weight vector is estimated for features extracted from this model; therefore it seems that it would be quite possible to have an arbitrary region size for the unsupervised model (perhaps not p).

Typo: line 256 ("taregt" -> "target")
Summary: This paper presents a semi-supervised extension of CNN for text categorization. Overall, this is a nice paper. The model is a natural semi-supervised extension of recent supervised efforts in this direction and the results are solid, if not amazing.

Submitted by Assigned_Reviewer_4

This paper presents a new semi-supervised CNN framework for text categorization. It can learn embedding of text regions with unlabeled data and then labeled data. Instead of using pre-trained word vectors, the model proposed in this paper tries to learn the predictiveness of co-presence and absence of words in a textual region.

With effective use of unlabeled data, the model has a better performances than the previous studies on sentiment classification and topic classification. Experimental results show that the model achieved the state-of-art accuracy.
Summary: This paper presents a new semi-supervised CNN framework for text categorization. It can learn embedding of text regions with unlabeled data and then labeled data.

Author Feedback
Author rebuttal: We would like to thank all the reviewers for valuable feedback. If the paper is accepted, we will improve our paper based on the reviews.